# Liquid Metal-Based Flexible and Wearable Sensor for Functional Human–Machine Interface

**DOI:** 10.3390/mi13091429

**Published:** 2022-08-29

**Authors:** Ye Tao, Feiyang Han, Changrui Shi, Ruizhe Yang, Yixing Chen, Yukun Ren

**Affiliations:** State Key Laboratory of Robotics and System, Harbin Institute of Technology, West Da-Zhi Street 92, Harbin 150001, China

**Keywords:** flexible sensor, PDMS composite, detection of bending angle, gesture-control

## Abstract

Rigid sensors are a mature type of sensor, but their poor deformation and flexibility limit their application range. The appearance and development of flexible sensors provide an opportunity to solve this problem. In this paper, a resistive flexible sensor utilizes gallium−based liquid metal (eutectic gallium indium alloy, EGaIn) and poly(dimethylsiloxane) (PDMS) and is fabricated using an injecting thin−line patterning technique based on soft lithography. Combining the scalable fabrication process and unique wire−shaped liquid metal design enables sensitive multifunctional measurement under stretching and bending loads. Furthermore, the flexible sensor is combined with the glove to demonstrate the application of the wearable sensor glove in the detection of finger joint angle and gesture control, which offers the ability of integration and multifunctional sensing of all−soft wearable physical microsystems for human–machine interfaces. It shows its application potential in medical rehabilitation, intelligent control, and so on.

## 1. Introduction

In recent years, wearable and flexible sensors [1,2,3] have attracted much attention because of their potential applications in soft robotics [4], health monitoring [5], and flexible electronics [6]. Compared to conventional sensors based on rigid materials and structures [7,8,9], soft wearable sensors worn on robotic or human hands or skin fabricated with flexible or elastomeric materials can convert mechanical stimuli into electrical signals [10]. Owing to the advantages of lightweight, flexible, stretchable, and biocompatible characteristics [11], the soft wearable sensors capable of sensing various signals such as temperature, contact force, pressure, and piezo−conductivity can be used in a variety of applications ranging from health monitoring systems and personal diagnostics to human–machine interfaces [12,13,14].

In contrast to these composite materials, liquid metals are metals or alloys that have the properties of both liquids and metals [15]. Gallium-based liquid metal, including eutectic gallium indium alloy (EGaIn, 75 wt% gallium, 25 wt% indium) [16,17,18] and gallium indium tin alloy (Galinstan, 68.5 wt% gallium, 21.5 wt% indium, 10 wt% tin) as examples, are commercially available and used widely in the literature [19,20,21]. The unique features of liquid metals have been employed for fabricating miniaturized microfluidic components including pumps, valves, heaters, and electrodes. In addition, Ga−based liquid metals are biocompatible and have chemical stability and, thus, the potential for application in intelligent wearable devices, biosensors [22,23,24,25], biomedicine, and skin therapy. However, studying and applying liquid metal-based flexible and wearable sensors within the context of microfluidics is a relatively new field of research and is evolving very quickly [26,27,28]. Thus, taking advantage of microfluidics technology or structures for patterning and embedding the liquid metal, a design of flexible and wearable sensors is utilized in this study for the development of a functional human–machine interface [29,30,31].

In the present paper, a long service life, wearable functional human–machine interface with integrated electrode sensors is developed [32], as shown in Figure 1a. We embed conductive eutectic gallium indium alloy and PDMS as materials featured with injecting the micron−scale microchannels [33,34,35]. The unique wire−shaped electrode design enables sensitive resistance detection under tensile, tactile, and bending loads (Figure 1b) [36,37]. The result exhibits high linearity of 98.29% and 97%, and a minimal hysteresis error of 3% and 4%, respectively. The wearable sensor also demonstrates excellent robustness over 10 repeat cycles of multidirectional stretching and bending and a minimal repeatability error less than 4% (Figure 1c), and subsequent experiments show that it also has good fatigue resistance. For the functional human–machine interface, the sensor is combined with gloves and a wireless module [38,39,40] to measure finger bending angle very sensitively (Figure 1d,e), and a series of applications has been implemented such as controlling the Bluetooth car with two−finger gestures with multidirectional stretching and bending (Figure 1f) [41]. The results demonstrate the multifunctional sensing abilities of the developed liquid metal−based sensor and verify its potential in wearable applications as an interaction interface between humans and machines.

## 2. Materials and Methods

### 2.1. Flexible Substrate Material

Firstly, the material selection of the flexible sensor (including substrate material and sensitive material) was studied. The physical properties of sensors, such as formability, flexibility, and resistance to destruction, are mainly determined by the flexible substrate materials, so the selection of appropriate materials is an important part of sensor preparation. Commonly used flexible substrate materials include polydimethylsiloxane (PDMS), silica gel, polyethylene terephthalate (PET), polyimide (PI), and so on. Among all flexible materials, PDMS has the characteristics of adjustable physical properties, transparency and colorlessness, good thermal stability, and good biocompatibility. Therefore, PDMS was selected as the body material of the flexible sensor. Next, the relationship between the physical properties of PDMS and the ratio of fabricated materials was investigated, the most important of which are elastic modulus and maximum tensile strain. The physical properties of PDMS are mainly related to the ratio of the curing agent and silicone oil. The PDMS material was molded into standard samples, applied loads, and measured mechanical properties by an electronic universal testing machine (Appendix A). The force model of the sample is shown in Figure 2a,b.

The data measured by the test machine are the load *F* and displacement *x* during tension or compression. The nominal strain $\varepsilon_{m}$ and nominal stress $\sigma_{m}$ are calculated as shown in Equations (1) and (2), while real strain $\varepsilon_{z}$ and real stress $\sigma_{z}$ are calculated by Equations (3) and (4). Through the measurement system of the experimental machine, the stress−strain curve is drawn by combining Equations (1)−(4), and the elastic modulus E of the flexible material is calculated by Equation (5).
(1)$$\varepsilon_{m}=\frac{x}{l_{0}}$$
(2)$$\sigma_{m}=\frac{F}{A_{0}}$$
(3)$$\varepsilon_{z}=\int^{\;}\frac{dl}{l}=\ln l-lnl_{0}=\ln(1+\varepsilon_{m})$$
(4)$$\sigma_{z}=\sigma_{m}\times \left(1+\varepsilon_{m}\right)$$
(5)$$E=\frac{\sigma }{\varepsilon }$$
where $l_{0}$ denotes the initial length of the sample and $A_{0}$ denotes the cross−sectional area of the sample center before loading. In this paper, $l_{0}$ = 15 mm, $A_{0}$ = 14 ${\mathrm{mm}}^{2}$. The stretch and compressive stress−strain data are shown in Figure 2c,d. The elastic modulus of the material under small deformation ($\varepsilon_{z}$ = 0.2) is summarized in Appendix A. As can be seen from the diagram, the larger the proportion of the curing agent, the higher the Young’s modulus of the material in tensile and compression.

After that, the effect of silicone oil content on the elastic modulus of PDMS was analyzed. Different proportions of silicone oil (viscosity 50) were added to PDMS with curing agent ratio of 10:1. The stress−strain relationship diagram was obtained (Figure 2e,f). The elastic modulus of the material at small deformation ($\varepsilon_{z}$ = 0.2) was summarized in Appendix A. It can be seen that the higher the proportion of silicone oil, the higher the Young’s modulus of tensile and compressive materials.

Based on the above information, reducing the curing agent or increasing the silicone oil can reduce the elastic modulus of PDMS. However, in the experiment, it was found that when too much silicone oil was added (the ratio is greater than 1:4), the crack sensitivity of PDMS materials increased, and it easily fractured in the tensile process. Therefore, the ratio of substrate, curing agent, and silicone oil was 20:1:2.

### 2.2. Fabrication of Flexible Sensor

In the field of microfluidics research, soft lithography is often used to fabricate the fluid channel of the microfluidic chip, and the height and width of the channel can be processed to the micron level. Accordingly, the liquid metal channel of the sensor can be fabricated in this way. The production process is as follows: (1) Photoengraving mask making. (2) Stick Dupont membrane. (3) Lithography. (4) Development. (5) PDMS pouring. (6) Bonding. (7) Liquid metal injection. (8) Connecting the wire and sealing. The specific method and process are shown in Appendix A as well as Appendix A.

### 2.3. Method of Measuring Resistance

After the sensor was fabricated, a multimeter was used to measure the sensor, and its initial resistance value was $R_{0}$. Equation (6) can be used to calculate the theoretical resistance value of the sensor.
(6)$$R=\rho \frac{L}{S}$$
where ρ denotes the resistivity, L denotes the resistance wire length, and *S* denotes the cross-sectional area of the resistance wire. The electrical conductivity σ of liquid metal is $3.4\times 10^{6}\;S/m$; therefore, the electrical resistivity ρ is $2.94\times 10^{-7}\Omega \cdot m$. The information to be collected in this paper is the resistance value. Based on the principle of resistance voltage division, an Arduino development board was used to build an external signal acquisition circuit, and programs were written to receive and process data. The principle of resistance measurement was shown in Appendix A. One end of the sensor resistor $R_{x}$ was grounded, the other end was connected in series with a known resistor $R_{0}$, and the other end of the resistor $R_{0}$ was connected with a 5 V power supply $V_{in}$. $V_{out}$ is the voltage between resistors $R_{0}$ and $R_{x}$. After connecting to the development board, this port can collect voltage signals without conversion. The resistance of the sensor is calculated by Equation (7).
(7)$$R_{x}=\frac{V_{out}}{V_{in}-V_{out}}\cdot R_{0}$$

## 3. Results and Discussion

### 3.1. Mechanical Performance Test of the Sensor

We planned to apply different loads to the flexible sensor and designed the corresponding measurement system to test its performance. For example, stretch and bending loads were applied to the linear sensor respectively, and the deformation data were collected, and then compared and analyzed with the theoretical value after processing. We built a resistance measurement platform that needs to include three parts: (1) the loading platform, where the sensors applied tensile and bending load; (2) the signal collection circuit, where, first, the resistance of the sensor signal can be converted to electrical signals and then converted into a digital signal and finally to the computer; and (3) data reception procedures, which receive and send the data acquisition circuit processing.

The linear sensor can be applied in the deformation or displacement measurement along the channel direction. The tensile test platform is shown in Appendix A. Two flat clamps are first used to secure both ends of the sensor (Figure 3b), ensuring that the direction of the channel and the direction of the tensile load are the same. Then the clamping distance between the two ends of the clamp is measured, that is, the initial length of the sensor. Finally, the deformation data of the sensor is obtained by programming. To attach the linear sensor to the knuckle of the hand to measure the bending angle, another bending test device is designed. Based on the idea of simplifying the finger joint as a hinge, the designed three−dimensional model of the test device is shown in Figure 3a. The object made by the 3D printing method is shown in Appendix A.

The sensor was stretched one-way at a speed of 1 mm/s, and the data obtained are shown in Figure 3c. It can be observed from the figure that the resistance and deformation of the sensor change almost linearly. When the deformation reaches the maximum value of 20 mm, the elongation is 74%, the maximum resistance is 49.2 Ω, the growth rate of resistance is 111.959%, and the slope of the linear fitting is 1.344 Ω/mm. The calculation of nonlinearity is shown in Equation (8).
(8)$$r_{L}=\pm \frac{\Delta L_{max}}{y_{FS}}\times 100\%$$
where $\Delta L_{max}$ denotes the maximum deviation of linear fitting and $y_{FS}$ denotes output value at full scale. In this paper, $\Delta L_{max}$ = 0.444 Ω, $y_{FS}$ = 26.0 Ω and the nonlinearity is 1.71%, and the sensor shows very good linearity in the stretch along the channel direction.

The sensor was tested by reciprocating stretching twice, and the experimental conditions were consistent with one−way stretching. The data obtained are shown in Figure 3e. From the above figure, it is observed that the resistance variation of the sensor is still linearly related to the stretch amount, and the curves of the two stretching processes almost completely coincide, indicating high repeatability; the non−coincidence degree is calculated by Equation (9). However, hysteresis error exists in the stretching process and recovery process in the same reciprocating process, and the possible reasons are as follows: (1) For any kind of metal material, because of the complexity of its internal structure relationship, under the pressure of external force, microstrain will be generated between the tiny grains. After the external force disappears, the microstrain will disappear, but whether it completely disappears and returns to the original shape state depends, i.e., different materials have completely different behaviors. (2) Direct contact between liquid metal and PDMS has adhesion. When PDMS is deformed, liquid metal adhering to the inner surface of PDMS will not flow with the deformation of the channel, which also leads to hysteresis to a certain extent.
(9)$$r_{H}=\pm \frac{1}{2}\frac{\Delta H_{max}}{y_{FS}}\times 100\%$$
where $\Delta H_{max}$ denotes maximum return error. In this paper, $\Delta H_{max}$ = 1.26 Ω, $y_{FS}$ = 24.51 Ω, and the non−coincidence $r_{H}$ = 2.57%, indicating that the sensor has minimal hysteresis.

Next, the performance of the linear sensor was tested when bending from 0° to 90°. Three experimental devices with different bending diameters (9, 12, and 20 mm) were used to simulate three joints of different thicknesses of the index finger. Control the steering gear to rotate from 0° to 90° so that the bending of the linear sensor traverses every angle, during which the change of sensor resistance signal is collected. Linear and quadratic fitting were performed on the data, and the fitting results obtained are shown in Figure 3e.

After further analysis and summary of the data, Appendix A is obtained, and the following observations can be obtained:(1) The R−square of the linear fitting is above 0.99, and the nonlinearity is about 3%. The coefficient of the quadratic term of the quadratic fitting is about $1\times 10^{-3}$, which is much smaller than the coefficient of the primary term. It can be concluded that the resistance value of the linear sensor under bending load has a good linear relationship with the bending angle. (2) The non−coincidence degree of the sensor is about 3%, so the hysteresis of the sensor is small.

Finally, the performance of the sensor in reciprocating bending was tested. The bending period is 2 s, and the bending angle ranges from 0° to 100°. After repeated bending more than 10 times, the data obtained are shown in Figure 3f. As can be seen in the figure above, the resistance signals during repeated bending almost coincide. The non−repetition degree of the sensor can be calculated by Equation (10) (Appendix A).
(10)$$r_{R}=\pm \frac{\Delta R_{max}}{y_{FS}}\times 100\%$$
where $\Delta R_{max}$ denotes the maximum signal difference of multiple sensor operations. It can be seen from the figure that when the bending diameter is 9 mm, $\Delta R_{max}$ = 0.2421 Ω,$\;y_{FS}$ = 12.3817 Ω, $r_{R}$ = 1.955%; When the bending diameter is 12 mm, $\Delta R_{max}$ = 0.4836 Ω, $y_{FS}$ = 15.1074 Ω, $r_{R}$ = 3.201%. The repeatability of the sensor is not more than 4% in 10 times of bending, indicating that the sensor has good repeatability.

### 3.2. Construction of the Sensing Glove System

According to the previous conclusion, the bending of the sensor at the hinge joint is linear, so a flexible sensor can be configured on the hand to measure the bending angle of the finger joint. We improved the channel structure of the flexible sensor, designed the PCB circuit board for signal acquisition, and wrote the data processing program to design a wireless sensing glove (Figure 4b).

There are three joints in the human index finger and middle finger. To ensure that the measurement covers all joints and reduces the number of sensors, a single sensor with the same number of joints is directly arranged on one finger. The structure of the flexible sensor is redesigned. Figure 4a shows the index finger flexible sensor. Linear channels of different lengths are set at the three joints to adapt to the bending radius of different joints. By connecting the three channels inside the sensor, the number of external wires is reduced from six to four, and liquid metal can be injected into all channels at once. The sensor on the thumb is a single−joint sensor, and on the index and middle fingers are three−joint sensors, as shown in Appendix A.

The working principle of signal acquisition is shown in Figure 4d. The comparison resistor $R_{0}$ and all sensors $R_{1}$~$R_{9}$ on the glove are connected in series as a circuit, and 2–10 mA DC is passed. The STM32 single chip microcomputer has 10 signal input ports $A_{0}$~$A_{9}$, one end of the comparison resistor $R_{0}$ is grounded, from the other end of $R_{0}$ to connect to port $A_{0}$, port $A_{1}$, port $A_{2}$ … Finally, the sensor $R_{9}$ connects to port $A_{9}$. The potential $V_{0}$~$V_{9}$ of 10 resistors can be detected. On one end of the resistor, $R_{0}$ is grounded, so the partial voltage of $R_{0}$ is the potential of $A_{0}$ port: $U_{0}$ = $V_{0}$. The partial voltage of other resistors is the voltage between adjacent ports: $U_{n}$ = $V_{n}-V_{n-1}$. Since all sensors and comparison resistors pass the same DC $I_{0}$, Equations (11) and (12) can be obtained.
(11)$$I_{0}=\frac{U_{0}}{R_{0}}=\frac{U_{1}}{R_{1}}=\ldots =\frac{U_{n}}{R_{n}}$$
(12)$$R_{n}=\frac{U_{n}}{U_{0}}\cdot R_{0}=\frac{V_{n}-V_{n-1}}{U_{0}}\cdot R_{0}$$
(13)$$k=\frac{\theta_{a}-\theta_{b}}{R_{b}-R_{a}}$$
(14)$$\theta =\theta_{a}+k\cdot \left(R-R_{a}\right)$$

When a linear sensor is bent, its resistance value is linear with the bending angle. The diameter of the bending joint determines the slope *k*. Therefore, the slope *k* when the sensor is working can be obtained only by obtaining the angle $\theta_{a}$ and $\theta_{b}$ at any two points in the bending process and combining the resistance $R_{a}$ and $R_{b}$ (Equation (13)). The joint angle θ is calculated as shown in Equation (14). The signal is transmitted to the computer through the Bluetooth module, and finally, the data is processed and analyzed by the program in the software.

### 3.3. Check the Bending Angle of the Finger Joints

The single−joint sensor on the thumb and three-joint sensors on the index finger are used as experimental objects to test the bending performance. The angle detection ability of the sensor glove on the second joint of the thumb was tested by using the single−joint sensor at the thumb, as shown in Appendix A. The process of gesture operation is shown in Figure 5a: First, calibrate the joints at 0° and 90°, then bend them at 0°, 30°, 60°, and 90° in sequence, and hold them for about 4 s, as shown in sections G1, G2, G3, and G4 in Figure 5b. Then the bending process of 90−0° and 0−90° was realized at uniform speed within 5 s, as shown in sections G5 and G6. It can be seen from the observation in the figure that the calculated data of the sensor gloves are basically consistent with the actual bending angle. We further processed the data to obtain information such as measurement errors. The analysis shows that the average angle detected by the sensor gloves is generally greater than the target value, and the average error varies from 0.5° to 3.5° for different angles. The error range is larger, ranging in length from 12° to 20°.

Appendix A shows the layout of joint sensors on the index finger, and three sensors correspond to three joints. We performed a similar experiment on the thumb, measuring the angle detection ability of the sensor on the index finger. Based on the theory above, there is a certain linear relationship between resistance value and bending angle (Figure 3c,e), so different gestures change the resistance of the sensor. The starting angle $\theta_{a}$ = 0°and the ending angle $\theta_{b}$ = 90° of the sensor is set, and the calibration is completed by two actions of expanding the palm and clenching the fist, and the starting point calibration resistance $R_{a}$ and the ending point calibration resistance $R_{b}$ of each sensor are saved. After the calculation of the program, the fitting slope *k* of each sensor is obtained, and the current real−time joint angle is calculated according to Equation (14) and the resistance data.

Figure 5c shows all actions in the process of gesture operation, and Figure 5d shows the angle data collected corresponding to the above actions. According to the data in Figure 5d, we can intuitively see the motion of each joint. When the data of each sensor is observed separately, its characteristics are the same as the various characteristics of the thumb joint sensor data, but when one joint is bent, the other knuckles produce a slight implicated movement, so the sensor signal of the inactive joint fluctuates slightly.

We analyzed the possible causes of the errors: First of all, when the sensor gloves are calibrated at 0° and 90°, the finger joints cannot be accurately bent in place, so the calibration steps cause errors. Secondly, the process of joint operation experiment is highly subjective, and it is virtually impossible to ensure that each joint can be operated accurately according to the target angle curve in Figure 5a,c, which is another major source of errors.

### 3.4. Gestures Control the Motion of the Car

To better display the function of the sensor glove, we applied it to the motion control of a Bluetooth wireless car and planned to realize its motion control through different gestures. Considering the limitations of the first joint of the middle finger and index finger, we used the single joint sensor on the thumb and the double joint sensor on the index and middle finger to carry out the application experiment. The specific steering control mode is shown in Appendix A.

In practice, the Bluetooth car could realize the desired movement according to different gestures. Next, we carried out some general operations, and the following actions are demonstrated in Appendix A:

Straight forward: Set the motion direction of both wheels as forward, then the third joint of the index finger and middle finger kept the same angle, gradually bent from 0° to 90°; the car could keep straight forward, and with the increase of finger bending angle, the car could achieve a 0−5 speed increase, as shown in Figure 6a and Appendix A.

Motion direction switch: When the car is stationary or moving, and the second joint of the index finger or middle finger is quickly bent at the same time, the car can realize the tire motion direction switch within about a 500 ms delay. For example, when the car quickly completes this action in the process of straight forward, the car can immediately switch the movement direction to achieve a straight retreat.

Differential steering: Foremost, set the same direction of movement of the two wheels. When the bending angle of the third joint of the two fingers is more than 15−20 degrees different, the two wheels of the car carry out differential steering because of the different speed levels. This remote−control glove uses the right hand, and the index finger is located on the left side of the middle finger. When the left index finger is bent at a larger angle, the car turns to the left. Otherwise, it turns to the right. This is more consistent with common people’s thinking about remote-control habits. The greater the difference in bending angle, the greater the difference between the two wheels, and the car steering action occurs more quickly (Figure 6b).

Rotation in place: In the beginning, switch the direction of the two wheels so that they turn in opposite directions. When the speed level of the two wheels is the same, the two wheels can be realized for in situ clockwise or counterclockwise rotation. When the speed level is different, in addition to the rotation of the car, its center position also changes significantly, as shown in Figure 6c.

## 4. Conclusions

In summary, we innovatively added silicone oil to the sensor body by using the soft lithography method, combined with the resistance characteristics of liquid metal, and determined the configuration ratio of substrate, curing agent, and silicone oil to be 20:1:2 to obtain a suitable elastic modulus and successfully produced a flexible sensor. After stretching and bending experiments, the linearity of the sensor is about 1.7% when stretching and 3% when bending. The degree of non−repetition is about 3%, which shows that it has good linearity and repeatability. Liquid metal sensors have the advantages of flexible characteristics, strong plasticity, excellent electrical conductivity, and so on, which have broad application prospects in the field of biology and human−computer interaction. This paper combines liquid metal flexible sensors with specific circuits and other devices to design and manufacture wireless sensing gloves, which are used to measure human finger joint movements, fingertip pressure, and other information. Using the detected human body information, the function of the finger remote−control Bluetooth car motion is realized. In the future, it can be used in the field of medical rehabilitation to monitor patient status, and it can also be applied to virtual reality technology to provide new ideas and schemes for intelligent control and remote surgery.

## Figures and Tables

**Figure 1 micromachines-13-01429-f001:**
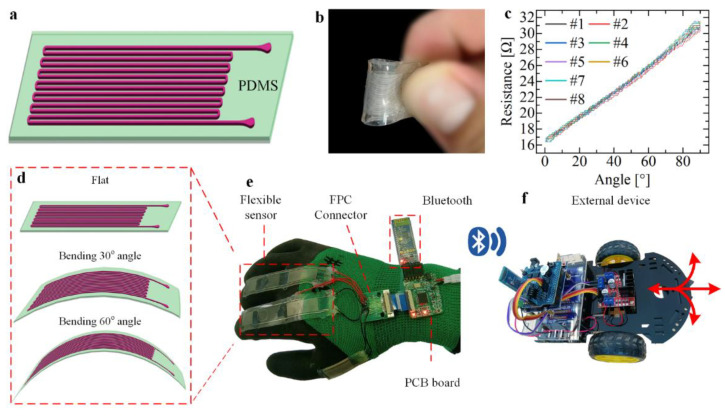
Preparation, performance research and application of the liquid metal flexible sensor. (**a**) Flexible sensor based on soft lithography. (**b**) Demonstration of bending of the sensor. (**c**) Performance test of linear sensor reciprocating bending; the bending radius is 6 mm. (**d**) Sensor-based deformation detection joint bending angle. (**e**) Flexible sensors are integrated into the glove. (**f**) Control wireless Bluetooth car based on sensing glove.

**Figure 2 micromachines-13-01429-f002:**
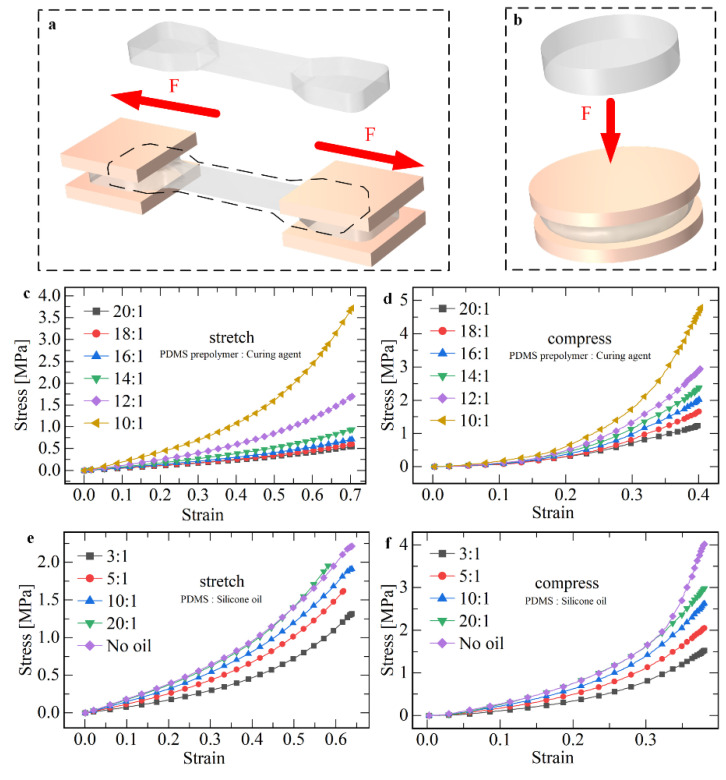
The effects of the content of curing agent and dimethyl silicone oil on the physical properties of PDMS are experimentally studied. The elastic modulus of the flexible material is tested and the stress-strain curve of PDMS is plotted. (**a**) Measuring the tensile properties of flexible materials. (**b**) Measuring the compressive properties of flexible materials. (**c**) Real tensile stress–strain curves of PDMS under different curing agent ratios. (**d**) Real compressive stress–strain curves of PDMS under different curing agent ratios. (**e**) Real tensile stress–strain curves of PDMS under different polydimethylsiloxane ratios. (**f**) Real compressive stress–strain curves of PDMS under different polydimethylsiloxane ratios.

**Figure 3 micromachines-13-01429-f003:**
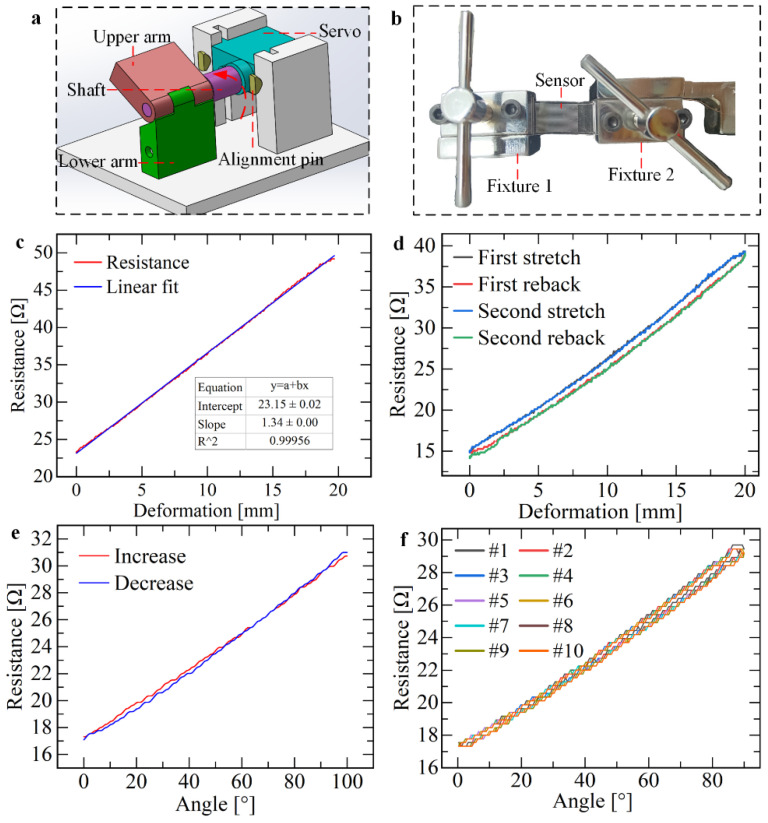
Mechanical properties test of sensors. (**a**) A 3D model of the test platform for measuring the bending angle. (**b**) Tensile measuring platform based on an electronic universal testing machine. (**c**) The one−way tensile resistance−strain curve of the sensor. (**d**) The reciprocating tensile resistance-strain curve of the sensor. (**e**) Point−by−point bending performance curve of the sensor. The bending diameter is 9 mm. (**f**) Reciprocating bending performance curve of the sensor.

**Figure 4 micromachines-13-01429-f004:**
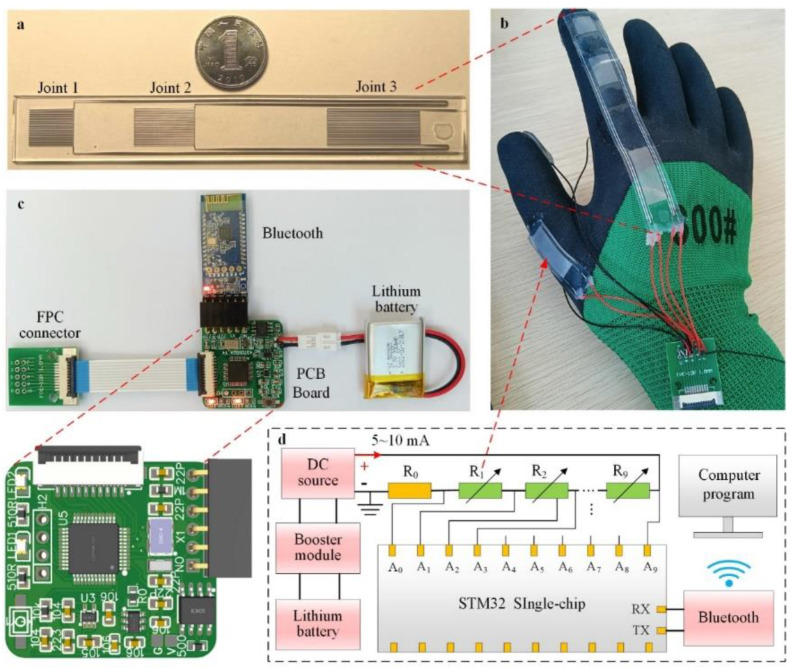
Design of the sensor glove and its signal processing system. (**a**) Sensors on the index finger joint. (**b**) Sensors are assembled in gloves. (**c**) The construction of the circuit control system of sensing gloves, and the enlarged part is the preview of the printed circuit board. (**d**) Schematic diagram of working principle of multi−resistance acquisition circuit.

**Figure 5 micromachines-13-01429-f005:**
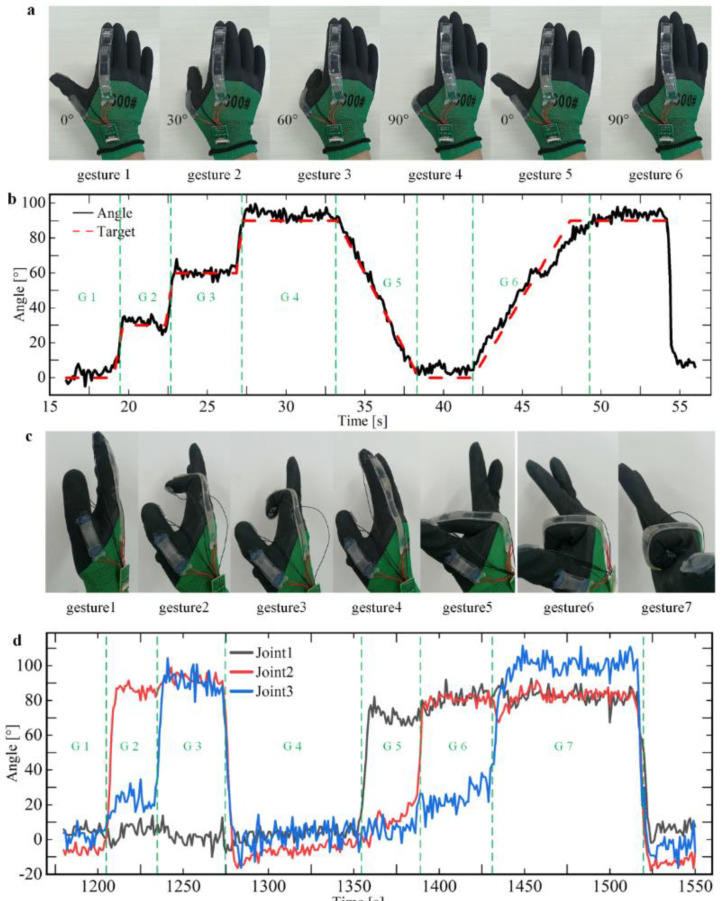
Testing and analysis of the bending performance of the sensing glove. The performance of the sensing glove refers to the ability to accurately detect the bending angle of the finger joint after placing the sensor on the glove and fitting it with the finger joint. (**a**) The gesture of bending one joint of the thumb, where the bending angle traverses 0−90°. (**b**) Detection of the bending angle of different gestures and test the degree of similarity between the measured angle and the target angle. (**c**) The gesture of bending the knuckles of the index and middle fingers at an angle from 0° to 90°. (**d**) Detection of the bending angle of multiple joints of the index finger.

**Figure 6 micromachines-13-01429-f006:**
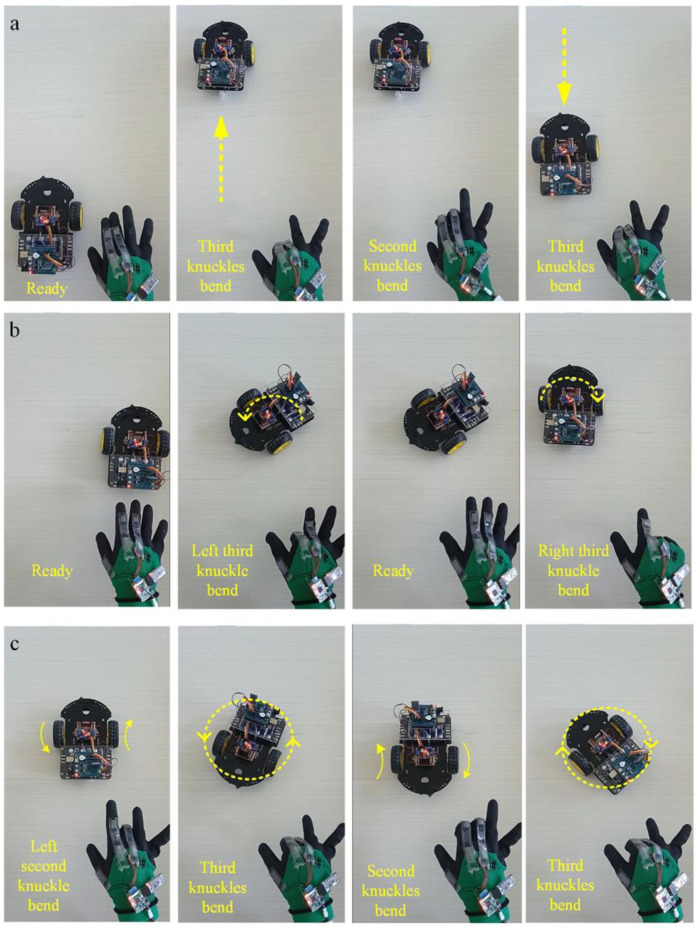
The sensor glove controls the motion of the Bluetooth car. (**a**) Gesture−control forward and backward for the wireless car. Change the direction of motion once. (**b**) Gesture−control left turn and right turn for the wireless car. (**c**) Gesture−control left turn and right turn for the wireless car in situ.

## Data Availability

Not applicable.

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
