# Peer review of "Liquid Metal-Based Flexible and Wearable Sensor for Functional Human–Machine Interface"

_micromachines, 2022, doi:10.3390/mi13091429_

Round 1
Reviewer 1 Report
The authors of the paper "Liquid metal based flexible and wearable sensor for functional human-machine interface" built a resistive flexible sensor out of gallium-based liquid metal and PDMS, which was created using an injecting thin-line patterning technique based on soft lithography. Combining the scalable fabrication process and unique wire-shaped liquid metal design enables sensitive multifunctional measurement under stretching and bending loads. Moreover, the flexible sensor is combined with the glove to demonstrate the application of the wearable sensor glove in the detection of finger joint angle and gesture control, which provides a path towards system-level all-soft and highly integrated wearable physical microsystems for human-machine interfaces. It shows its application potential in medical rehabilitation, intelligent control, and so on. The topic is within the scope of the journal. However, the manuscript needs major and sincere revision before being accepted.
1) Line 9, “Rigid sensor is a mature type of sensor”. It’s suggested to check the singular and plural forms in this sentence. (Example: “Rigid sensors are a mature type of sensor.”)
2) Line 12 & 13, It’s suggested to add an article before the word ‘injecting’; It will make the sentence more meaningful. (Suggestion: “using an injecting thin-line patterning technique”).
3) Line 15-18, Please rewrite this sentence. It should be simple and fluent for the readers. (Example: “Furthermore, the flexible sensor is combined with the glove to demonstrate the wearable sensor glove's application in the detection of finger joint angle and gesture control”).
4)Line 25, It’s preferable to use “Compared to conventional” instead of “Compared to the conventional”. (the)
5) Line 25-31, It’s strongly suggested that authors rewrite this sentence. (Suggestion: Focus on grammar and fluency.) This is a little confusing.
6) Line 58-60, (Figure1-C), Why do the authors think that it has "excellent fatigue resistance?" Please provide more information regarding this claim.
7) Line 84-85 (Figure2), “The elastic modulus of the flexible material was tested and the stress-strain curve of PDMS was plotted.” Please describe the testing method in more detail. It will be helpful for new readers. (Although all of the figures were appropriate and suitable).
8) It is recommended to use either "metal based" or "metal-based". Check the paper once again. (hyphen)
9) Figure5, A further explanation is required for the “performance of the sensing glove”. Authors are advised to explain a little more about “how it detects the bending angle of different gestures and tests the degree of similarity between the measured angle and the target angle.”
10) Line 172, Why hysteresis error exists in the stretching process? Authors are encouraged to elaborate on this matter.
11) An English language check is required for the entire manuscript. This will increase the quality of the manuscript.
Reviewer 2 Report
Authors have developed a flexible sensor by embedding a liquid metal in a synthetic polymer. This is for addressing the limitations of traditional electronic which are not stretchable and flexible. The manuscript is fairly written well and easy to understand. However, following are to be addressed.
1. It is suggested that the authors rewrite the following sentences in Abstract.
“Rigid sensor is a mature type of sensor”.
“which provides a path towards system-level all-soft and highly integrated wearable physical microsystem for human-machine interfaces”.
2. Fabrication process of the sensor is a very important information, and it is recommended that Figure S1 is moved to main manuscript.
3. Why have authors chosen only 10 repeat cycles? What are the maximum number of cycles until the functionality of the sensor is comprised?
4. What are the practical applications of the sensor? Author’s have used the sensor to operate a toy car. However, what are the potential applications in the areas of biosensing, biomedicine, and skin therapy?
Round 2
Reviewer 1 Report
The manuscript titled “Liquid metal-based flexible and wearable sensor for functional human-machine interface” is within the journal's scope. The manuscript is recommended for publication.